

# Reask UTC: a machine learning modeling framework to generate climate connected tropical cyclone event sets globally.

Thomas Loridan[1], Nicolas Bruneau[1]

[1]Reask, 70 Gracechurch Street, London, ECV3 0HR, United Kingdom

*Correspondence to*: Thomas Loridan (thomas@reask.earth)

**Abstract.**

In the early 1990s, the insurance industry pioneered the use of risk models to extrapolate Tropical Cyclone (TC) occurrence
and severity metrics beyond historical records. These probabilistic models rely on past data and statistical modelling techniques
to approximate landfall risk distributions. By design such models are best fit to portray risk under conditions consistent with
our historical experience. This poses a problem when trying to infer risk under a rapidly changing climate, or in regions where
we do not have a good record of historical experience. We here propose a solution to these challenges by rethinking the way
TC risk models are built, putting more emphasis on the role played by climate physics in conditioning the risk distributions.
The Unified Tropical Cyclone (UTC) modelling framework explicitly connects global climate data to TC activity and event
behaviours, leveraging both planetary scale signals and regional environment conditions to simulate synthetic TC events
globally. In this study we describe the UTC framework and highlight the role played by climate drivers in conditioning TC
risk distributions. We then show that, when driven by climate data representative of historical conditions, the UTC is able to
simulate a global view of risk consistent with historical experience. Additionally, the value of the UTC in quantifying the role
of climate variability on TC risk is illustrated using the 1980-2022 period as a benchmark.



## 1 Introduction

Tropical Cyclones (TCs) pose a threat to coastal communities across the globe. Recent examples include a record breaking 2022 season for Madagascar, where 5 storms made landfall causing up to 365 fatalities across Madagascar, Mozambique and

Malawi (Aon, 2022). Sadly, these regions were impacted again in 2023 by category 5 cyclone Freddy, causing up to three times more fatalities. From an economic stand point, TCs caused 92B USD of global economic losses in 2021, with hurricane Ida alone costing 75B USD (Aon, 2021) while in 2022 Category 5 hurricane Ian became the third costliest event on record with over 100B USD of economic losses (NOAA, 2023). At the time of writing, Hurricanes Helene and Milton have just hit the west coast of Florida with combined expected economic in excess of 50B USD (Morningstar, 2024).

A range of public and private organizations focus on mitigating this risk. To do so they require tools that quantify the occurrence and severity likelihood of events globally. Since the early 90s the insurance industry has adopted the use of large sets of synthetic TC events as a way to understand and quantify TC risk beyond simple analysis of historical records. These synthetic events all represent plausible TC scenarios, typically generated from statistical extrapolation of historical occurrences (Hall and Jewson, 2007; Rumpf et al., 2007, Vickery et al. 2009, Bloemendaal et al. 2020; Arthur 2021). The climatology and

statistics of such event sets (often referred to as stochastic event sets in reference to their generation process) are consistent with history, but allow extrapolation beyond what was observed. They help quantify probabilistic measures of risk such as the 1-in-100-yrs return period hazard intensity (i.e. an intensity level with a 1% annual chance of occurrence).

While such methods have greatly helped the industry better understand TC risk, they suffer from a fundamental limitation: they are mostly driven by statistics of past data rather than physics. At the core of the event generation process resides a series

of statistical relationships that are fit to historical data, and therefore best represent TC risk under conditions that are consistent with historical data points. This presents two important challenges when assessing global risk in a changing climate:

- A model anchored in past climate conditions is not able to adapt and quantify shifts in risks associated with a changing climate: e.g. how do TCs react to regional changes in patterns of dominant atmospheric steering flow, ocean temperatures or wind shear?

- A model fit to historical data will be best fit to those regions where we have abundant historical records (e.g. the North Atlantic) but will generalize poorly to other basins where data are scarce and TC behaviours may differ (e.g. the South Indian Ocean).

One solution to this problem is to build smarter event generation algorithms, that do not simply memorize and extrapolate history but also understand how climate physics influenced the observed outcomes. Explicitly linking the event generation

algorithms to key climate drivers allows the creation of *climate-connected* event sets that can naturally quantify risk (1) under changing climate conditions and (2) in regions where historical data are scarce. Several climate-connected TC event sets have recently been developed by the academic community, with leading modelling groups developing TC risk solutions that explicitly link some components of the event generation process to climate model outputs (Lee et al. 2018, Jing and Lin. 2020,



Emanuel 2021 and citations within, Lin et al. 2023, Sparks and Toumi 2024). We here present a novel approach (the Unified

Tropical Cyclone – UTC) that, while following a similar philosophy, differs in several key aspects:

- The UTC links the annual frequency of TC occurrence in each active basin to large scale environment signals (e.g. El Nino Southern Oscillation - ENSO) rather than through the use of more localized genesis potential indices (e.g. TCGI, see Wang and Murakami, 2020).


- The UTC directly simulates the impact of sea surface temperatures, atmospheric steering flow, mean sea level pressure and vertical wind shear on the TC trajectory and intensity hourly increment distributions thanks to a machine learning (ML) algorithm called quantile regression forest (Meinshausen 2006; Loridan et al. 2017, Lockwood et al. 2024, Bruneau et al. 2024). Using ML ensures the impact of local

environmental factors can be inferred directly from data without the need for any expert judgment in formulating or tuning the relationship.

- The UTC is initialized with reanalysis data and model simulations of the past (see the results of this study below), but also with seasonal forecast data and future climate projections (this will be the focus of a

follow up study). Figure 1 provides an introductory illustration of how the UTC risk distributions (here for annual major hurricane US landfalls) shift according to different climate forcing conditions. More details on this experiment are provided in section 3.



Major Hurricane US landfall / year

**Fig. 1. UTC modelled distributions of annual US major hurricane landfalls, under a range of climate forcing assumptions. Vertical dashed lines show observed levels of occurrence for each scenario.**

In this study we describe how climate gridded data are used to condition the UTC event generation algorithms: namely the TC occurrence frequencies by basin, genesis location, date, track trajectory, and intensity modules. We then show how such a climate connected approach can reproduce a risk climatology across the globe that is consistent with history, with minimal need for local tuning, track filtering or calibration. We then conclude by analysing the impact of climate variability on the UTC view of risk, considering alternative climates of the 1980-20222 period as forcing when deploying the model.




## 2 The Unified Tropical Cyclone (UTC) modelling framework

The UTC framework consists of a series of algorithms that allow generation of synthetic events from knowledge of global climate conditions (Fig. 2). By generating a large number (i.e. millions) of such climate-connected synthetic events, we aim to capture a complete view of TC risk under the climate conditions provided as input. An overview of the event generation framework is first provided in section 2.1. Section 2.2 details how the event generation algorithms are developed, combining reanalysis of past climate with historical TC event records. In section 2.3 we come back to the event generation framework

and formally list the sequence of algorithmic steps that make up the UTC.

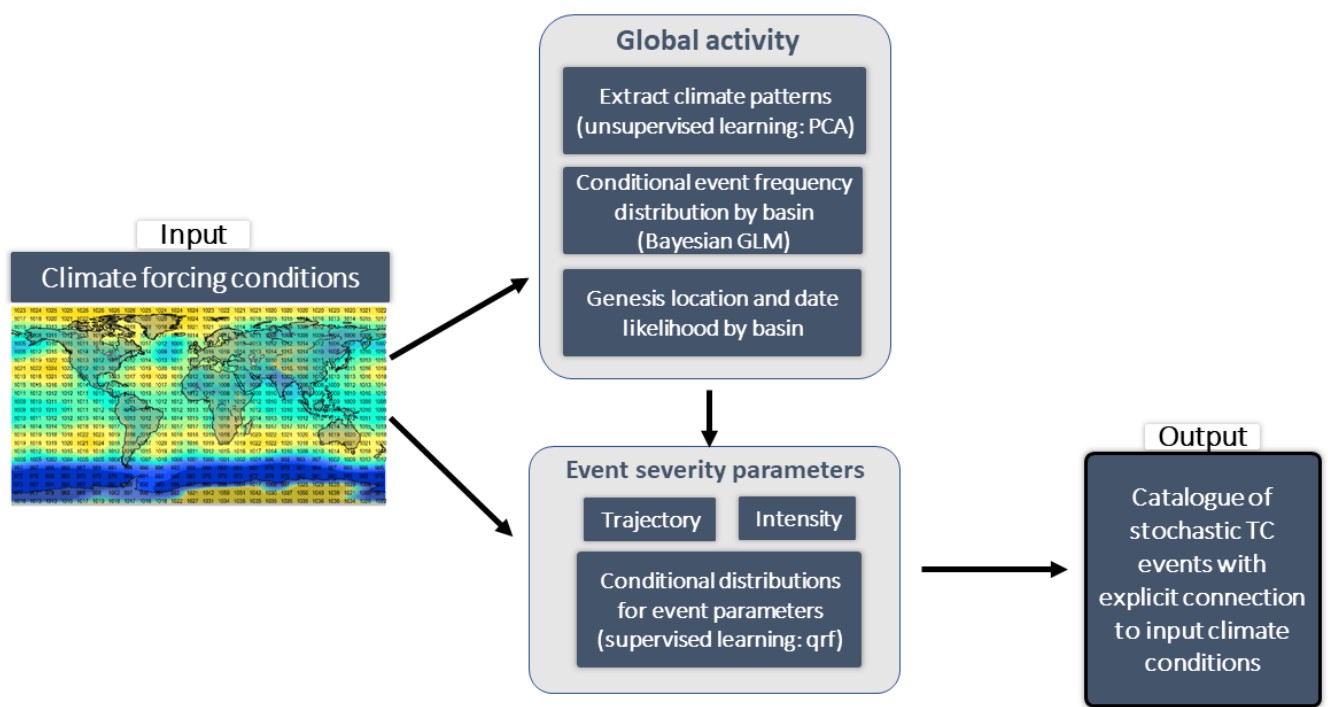

**Fig. 2. Overview of the UTC event generation framework. Global gridded climate data are used as input to a series of algorithms responsible for the generation of synthetic TC events. The output is a set of millions of events, representative of TC activity under**
**conditions set by the input climate data. Details of the algorithms are provided in section 2.2 and Appendix A.1.**

### 2.1 Event generation framework overview

The overarching objective when creating a *climate-connected* TC event set is to sample two dimensions of risk variability (see Fig. 1):

A.   The variability in TC risk under a given climate state (the distribution in each row of Fig. 1).

       B.   The variability in the climate state itself (different rows in Fig. 1).



While most traditional TC event sets are designed to address A (for a climate representative of a selected historical average state. e.g.. first row of Fig. 1), they fail to acknowledge the importance of B. To address B and drive the generation of events covering a wide range of climate conditions, we connect the UTC to global gridded climate inputs: ERA5 (Hersbach et al., 2020) is our preferred source to portray the climate experienced historically, and we also augment that view with alternative simulated climates from the CESM LENS2 project (Rodgers et al., 2021). While ERA5 data are available at higher resolution, we choose to aggregate to a similar 1-degree spatial resolution as the CESM LENS2 data. This choice is driven by a desire for consistency between model training (with ERA5) and deployment (using a wider range of sources not always available at the same resolution as ERA5, such as the CESM LENS2). When representing climate state at 1-degree spatial resolution we focus on variability in global and regional climate patterns rather than finer scale weather. In this framework the sampling of risk due to finer scale weather variability is part of dimension A above. This large database of monthly global gridded climate data is the starting point for our model deployment (Fig. 2). We limit the range of climate inputs to state variables that are known to impact TC dynamics:

- Sea Surface Temperature (SST)
- Mean Sea Level Pressure (MSLP)
- Zonal component of the wind flow at 850 mb (U850) and 200 mb (U200)
- Meridional component of the wind flow at 850 mb (V850) and 200 mb (V200)
- Vertical wind shear magnitude (SHR) – computed from the wind field components above
- Steering flow – also computed from the wind field components.

The UTC then implements the following modelling sequence (see Fig. 2):

1) A *climate state* is defined as a time series of monthly gridded climate data fields. From knowledge of key climate patterns in a given climate state (see section 2.2.1), the UTC models one distribution per basin to define event count likelihood during a TC season experiencing that climate (section 2.2.2). This process is repeated for a large number of climate states to capture dimension B described above.

2) Within each climate state, hundreds of different sample years (stochastic years) are computed to capture dimension A. These stochastic years account for variability in finer scale climate conditions (e.g. weather) not captured by the coarse climate forcing, as well as other stochastic TC behaviours occuring under a given climate state. For each stochastic year, and from knowledge of the distributions in 1, a number of events (*nTC*) per basin is sampled to define TC activity for that year.

3) For each of the nTC events sampled in a stochastic year, a likely genesis date and location are sampled from knowledge of historical occurrence rates and local environment conditions (section 2.2.3).

4) For each sampled event, the UTC simulates the trajectory of the TC centre at one-hour intervals, taking into account track persistence and the effect of environmental conditions such as the steering flow (section 2.2.4).



5)  Simultaneously, the evolution of the TC intensity (centre pressure) is also sampled along the track at one-hour
            intervals, from knowledge of the track characteristics to date and environmental conditions such as the vertical wind
            shear and ocean temperatures (section 2.2.5). An estimate of maximum sustained winds at 10 m is also computed
            from the modelled TC centre pressure following Bruneau et al. (2024).

By repeating the steps above for a large number of climate states (i.e. many years of climate forcing in 1) and a large number
of stochastic samples (i.e. repeated sampling of 2) the UTC generates a set of events characterizing risk variability across
dimensions A and B.

With complete record of the climate states used to generate any of the stochastic years, the UTC framework opens a whole
new range of analysis around the impact of climate variability (e.g. Fig. 1). By grouping years according to the phase of the El
Nino Southern Oscillation (ENSO), one can for instance quantify the resulting shifts in likelihood of TC landfalls across the
world, along with potential correlations between basins / regions. Similarly, questions around the impact of already realized
warming of the atmosphere on TC activity can be addressed objectively by sub-sampling the event set according to the warming
levels of the forcing climate states (e.g. first 3 rows of Fig. 1). From a risk analysis point of view, the UTC also helps identify
regions of the world that may have been lucky / unlucky in their historical experience compared to what should be expected
over the period of records (see section 3).


## 2.2 Event generation algorithms

When training the UTC event generation algorithms, we combine two data sources that jointly capture historical TC risk
conditions over the 1980-2020 period (i.e. the model *training period):*

- Monthly gridded data from the ERA5 reanalysis dataset (Hersbach et al., 2020) provide a best-estimate of the
155         climate experienced globally over the period.
- The International Best Track Archive for Climate Stewardship (IBTrACS, Knapp et al. 2010) records frequency,
            trajectory and intensity of TC events globally.

Most of the algorithms described below are based on machine learning (ML): i.e. derive their form directly from data rather
than from a human selected relationship. Throughout this section, key concepts are illustrated using case studies and simplified
algorithms where the physics is easily discussed. The complete algorithms, as implemented in the UTC, are detailed in
appendix A1.





### 2.2.1 Extracting patterns of climate variability impacting TC activity

The first step in the UTC framework is to reduce the dimension of the raw input of monthly gridded data into a selection of patterns important to TC activity. This dimension reduction phase is done via Principal Component Analysis (PCA, see appendix A1.1), performed on a range of standardized anomaly fields (FLD = SST, MSLP, U850, U200, V850, V200 and SHR - see section 2.1).

The end result for a given field is a series of spatial patterns ($PC_{FLD,i}$, see Fig. 3) allowing decomposition of any state of the

field using a series of coordinates ($W_{FLD,i}$, see equation A.1 in Appendix A1.1). While the PCA step provides important insights into the leading modes of global climate variability for each field (the $PC_{FLD,i}$), these are not all equally relevant to TC risk. To filter out the patterns most relevant to TC activity across different basins we rely on two criteria (Appendix A1.2):

- Only consider $PC_{FLD,i}$ modes whose weights ($W_{FLD,i}$) correlate with TC activity in at least one basin (candidate $PC_{FLD,i}$, see Fig. 4).

- Ensure the physical reasons for that correlation are understood. This is done by screening the patterns in the candidate $PC_{FLD,i}$ and explicitly linking them to conditions known to be favourable / unfavourable to TC genesis.

An example of the above is given in Fig. 3 with $PC_{SST,3}$ obtained from the SST decomposition of equation A1.1. The correlation between the number of North Atlantic hurricanes and the associated weights ($W_{SST,3}$), averaged over the July-November period,

is shown in Fig. 4. The reasons for the (negative) correlation between the magnitude of the weights and north Atlantic hurricane activity can be understood from analysis of Fig. 3: large values of the $W_{SST,3}$ weights are associated with anomalously warm SSTs in the eastern and centre Pacific (typical of an El Nino event) and anomalously cold SSTs in the tropical Atlantic. Both trends are signals of a likely weak hurricane season, which is confirmed by Fig. 4. Conversely, large negative values of the weight tend to be associated with La Nina type of Pacific SSTs and anomalously warm tropical Atlantic: i.e. conditions

favourable to hurricane activity.

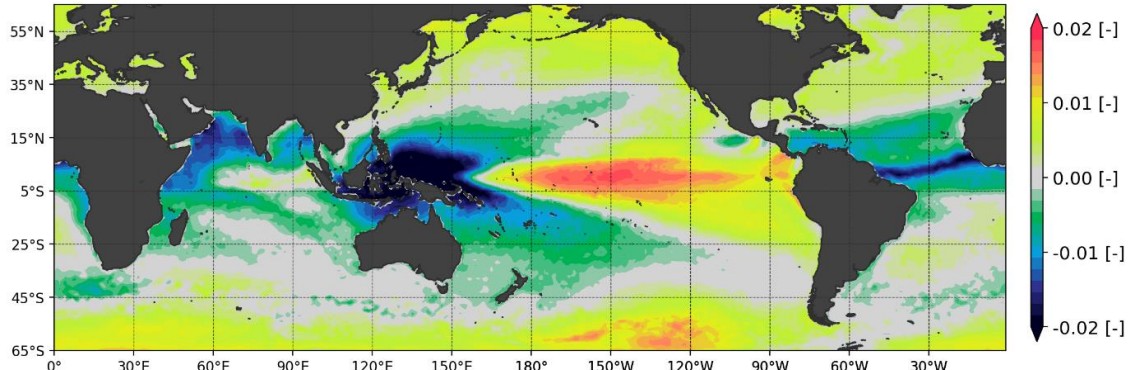

**Fig. 3: Principal component field representing one of the leading modes of global sea surface temperature variability ($PC_{SST,3}$). See equation A1.1 for details.**






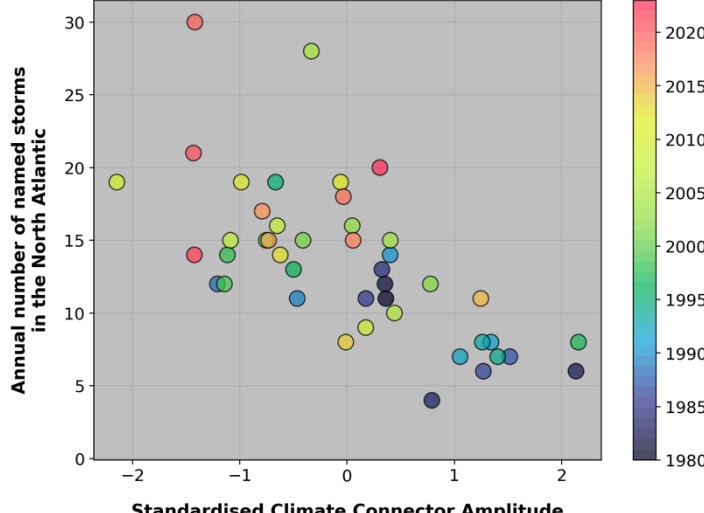

**Fig. 4: Relationship between the weight associated with the principal component from Fig. 3 (x-axis) averaged over the Jul-Nov period (i.e. $C_{NA,1}$, appendix A1.2) and the number of North Atlantic tropical storms in that season (y-axis). Colours indicate the season of record.**


Altogether a total of 13 patterns ($PC_{FLD,i}$) are selected to characterize climate states within the UTC framework (globally). As is the case in the example of Fig. 4 we maximize the correlation from the raw time series of weights by averaging over a time window that covers the peak TC activity period in each basin. The result is a set of 13 scalars that allow conditioning of TC activity in all active basins of the world. In what follows we refer to these scalars as *climate connectors* and the complete list

of connectors is provided in Appendix A1.2.

**2.2.2 Conditional distribution of TC numbers given magnitude of climate patterns**

By design the connectors selected in 2.2.1 correlate with TC activity in at least one basin. They therefore offer a way to link the input climate state to trends in basin-wide TC numbers. However, a large uncertainty exists around the exact number of TCs to expect under a given climate state (e.g. see vertical spread in Fig. 4 for a given $C_{NA,1}$ value).

Our approach to this challenge is to adopt a hierarchical Bayesian modelling framework, similar in concepts to that of Elsner and Jagger (2004). We use the connectors listed in Appendix A1.2 to condition the $\lambda$ rate of a Poisson distribution (see appendix A1.3). Fig. 5 illustrates the end result in a simplified case where the distribution of north Atlantic hurricanes is conditioned only on the value of the average July – November $W_{SST,3}$ shown in Fig. 4 (i.e. connector $C_{NA,1}$, see equation A2).

In years with large positive values of the connector (e.g. El-Nino years - see Fig. 3) the modelled distribution of hurricane

numbers shifts to a less active state (light blue), while for large negative connector values (e.g. La-Nina years) the shift is towards more frequent activity (red).





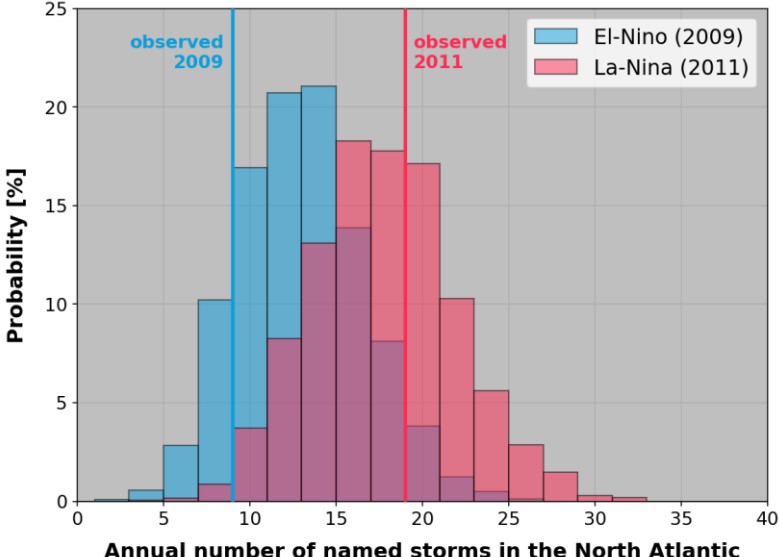

**Fig. 5: Modelled distribution of hurricane activity conditioned on the value of the average July – November W$_{SST,3}$ weight from Fig. 4, for 2009 (blue) and 2011 (red) climates. The vertical lines show observed activity levels in 2009 (El- Nino year) and 2011 (La-Nina year).**


For each basin that is TC active (i.e. North Atlantic, East Pacific, Western North Pacific, North Indian, South Indian and South Pacific basins) we have developed a different hierarchical Bayesian model using between 2 and 3 connectors. These are listed in Appendix A1.3. The ability of this approach to capture variability in TC basin frequency over the 1980-2022 period is illustrated in section 3.1 (see Fig. 10).


### 2.2.3 Genesis date and location within a basin

Once the level of activity in each basin has been established, the next step is to leverage patterns in both historical event occurrence and local environment conditions to determine the distributions of likely genesis location and date for all stochastic events. For these two components of the UTC we have so far relied on simple parametrizations rather than machine learning

methods. Upgrading this component of the system is a priority in future UTC development (e.g. following a Bayesian modelling approach as for section 2.2.2). In the interim we have implemented the parametrizations described below.

In a static TC risk model, the likely genesis location of a stochastic event is typically sampled from a probability density map representing a generalized version of historical records. An example of such a map is provided in Fig. 6a for the North Atlantic basin where a spatial smoothing was applied to all 1980 2020 genesis coordinates (i.e. 2D convolution using a 5x3 spatial

kernel). While this allows sampling from a climatology consistent with history, the approach does not account for season-to-season variability in climate conditions known to impact TC genesis likelihood. To condition the UTC genesis likelihood maps




we here adjust the static probabilities using a simple dimensionless scaling factor, that is based on the ratio of SST and SHR anomalies for each grid cell $k$.

$$scaling(k) = \frac{1 + {SST_{anomaly}(k)}\big/{|SST|}}{1 + {SHR_{anomaly}(k)}\big/{|SHR|}}$$

Physically, this adjustment ensures that the probability of genesis increases as the SST moves up from its climatological average and/or the vertical wind shear is reduced compared to its climatological state.

Fig. 6 provides an example of the adjusted genesis probability maps for two contrasting seasons: 2015 (Fig. 6b) and 1999 (Fig. 6c). In 2015, climate conditions show anomalously cold SSTs and strong wind shear conditions in the Caribbean Sea (Fig. 7a,c) while SSTs are anomalously warm in most of the mid-latitudes of the basin. This set up translates into an increased

likelihood of genesis along the US east coast and a reduction in the Caribbean Sea (Fig. 6b) when compared to the static historical baseline (Fig. 6a). Conversely, year 1999 is characterized by anomalously cold SSTs east of Florida and in the NW Gulf of Mexico (Fig. 7b) with very favourable shear conditions across the Caribbean and southern Gulf of Mexico (Fig. 7d). The impact on the modelled genesis likelihood map is towards an increased probability in the Caribbean Sea / Gulf of Mexico and a reduction east of Florida. In both years the patterns of actual observed event genesis (white circles, Fig. 6) are consistent

with these regional trends in favourable environmental conditions.

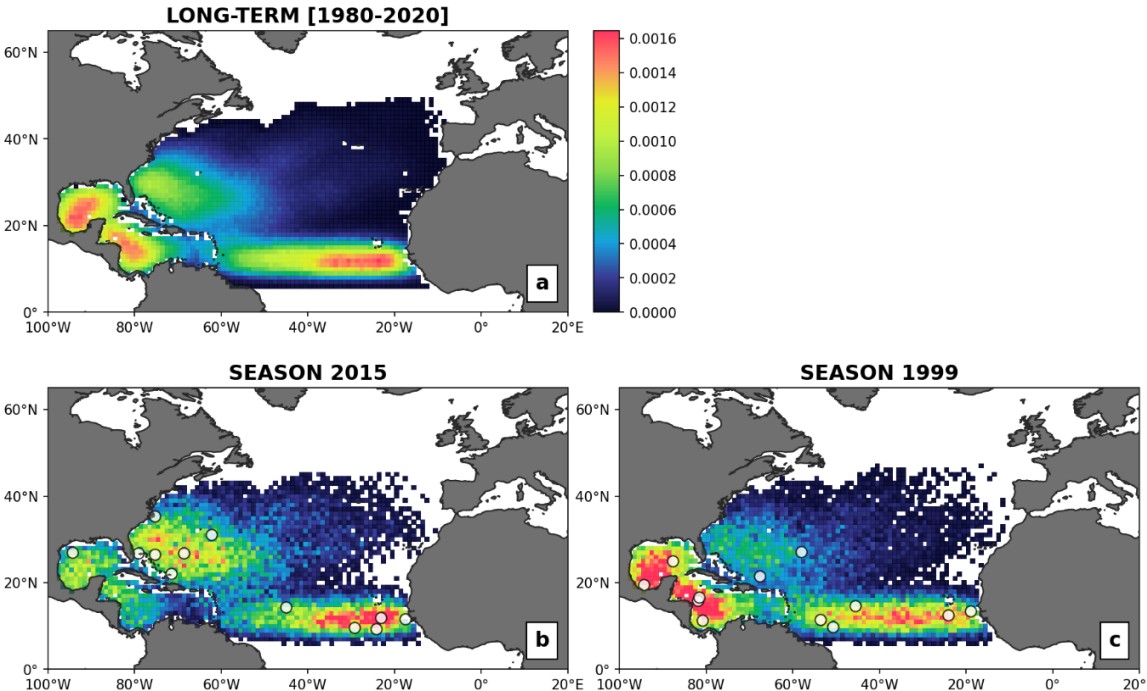

**Fig. 6: Spatial genesis density maps at 1 deg resolution; a) shows smoothed out static version of historical occurrences while b) and c) illustrate the dynamic spatial probability of genesis accounting for SST and wind shear anomalies. The white dots show the historical cyclone genesis occurrence for the two years considered (2015 and 1999, see Fig. 7).**



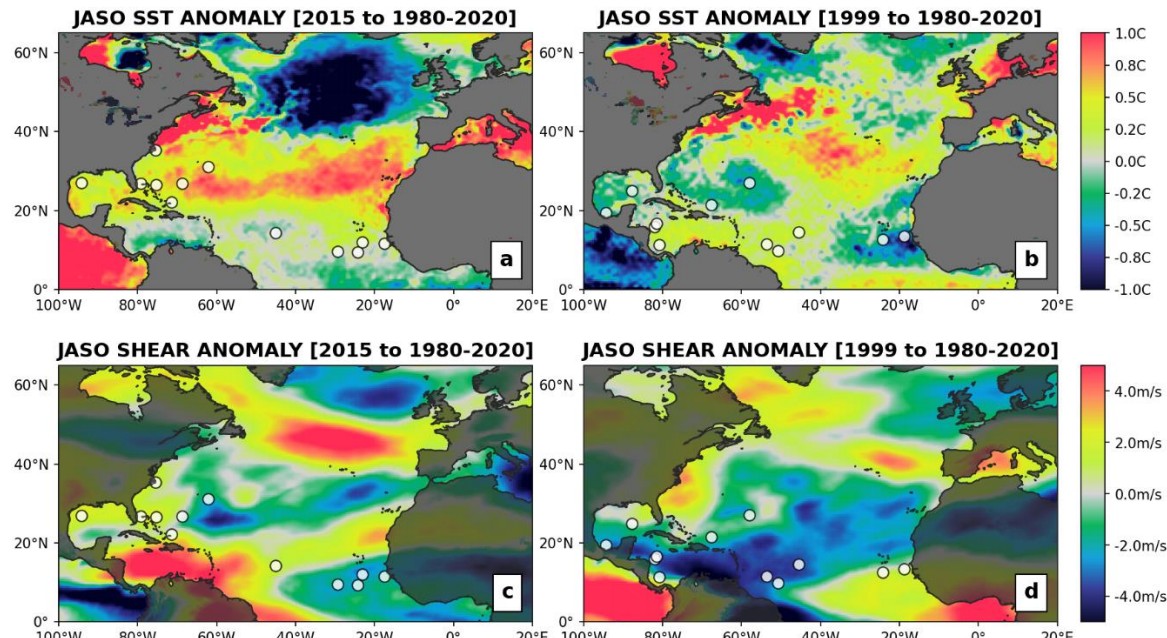

**Fig. 7: (a,b) SST and (c,d) SHR anomalies with regards to the 1980-2020 mean for July-October in year (a,c) 2015 and (b,d) 1999. Historical TC genesis locations are shown in white.**

Once the starting position of an event is known, a similar approach is used to allocate a starting date. The likelihood of genesis for a given month is computed as the average of two components:

- a probability density function fit to observed historical records ($P_{hist}$),
- a probability density function derived from monthly gridded SST, SHR and MSLP variables ($P_{clim}$).

Using historical genesis locations and associated climate conditions, three probability density functions ($P_{SST}$, $P_{SHR}$, $P_{MSLP}$) are first independently derived to link the likelihood of genesis in a month to different levels of monthly SST, SHR and MSLP. These are then combined into $P_{clim}$ as follows:

$$P_{clm} = P_{SST} \cdot P_{SHR}^2 \cdot P_{MSLP}$$

$P_{clim}$ allows conversion of the gridded climate fields into a monthly time series of probability maps. From knowledge of the sampled genesis location (see above), a time series of monthly probability is extracted and averaged with the climatological probability ($P_{hist}$). After linear interpolation of the monthly probabilities to daily resolution, a genesis day is sampled. Finally, the hour of genesis is uniformly sampled within the chosen day, and the date gets incrementally updated with the storm hourly displacements.



### 2.2.4 Track trajectory

As is common for most TC risk modelling systems (Hall and Jewson 2007, Blomenthaal et al. 2020, Arthur 2021), our approach to modelling individual event trajectories is to simulate incremental changes in latitude (*dlat* in deg/h) and longitude (*dlon* in deg/h) at fixed time intervals (1h in this study). Under that framework a track trajectory is simulated by iteratively sampling the next displacement from distributions conditioned on parameters at current and past locations (i.e. a Markov Chain Monte Carlo (MCMC) approach).

However, instead of relying purely on the track history to date and its location to predict the next *dlat* and *dlon* increment distributions, the UTC algorithms are also trained to account for local environmental conditions capturing the dominant steering flow. To do so we have overlaid the ERA5 reanalysis dataset on top of all historical TC events as reported in IBTrACS, and have trained a quantile regression forest algorithm (see Appendix A1.4) to approximate the *dlat* and *dlon* distributions, conditional on regional steering patterns. At any time step along the track the algorithm takes the following quantities as input

to condition the distributions: storm translational speed, track heading angle as well as incremental changes in latitude, longitude since previous time step, meridional and zonal components of the steering flow and spatial gradients of wind shear, mean sea level pressure and sea surface temperature. To ensure the algorithms generalize information globally rather than memorize local historical behaviours, no direct information of location (lat, lon coordinates) are provided. Conditional on this information, a distribution of *dlat* and *dlon* is modelled at every time step allowing sampling of the next hourly track

displacement (see section 2.3).

Fig. 8a illustrates the role played by environmental conditions in simulating event trajectories in the UTC. Using a 20 000 years subset form the event set of section 3.1, we extract the sampled *dlon* values of all events that pass through a selected region of the North Atlantic mid-latitudes (see map in Fig 8a). Sampled *dlon* values that correspond to time steps when monthly steering winds are predominantly blowing east are shown as a red distribution while the blue distribution represents

time steps with steering winds blowing west. The ability of the UTC to react to dominant steering patterns is clear from the shift between both *dlon* distributions, with simulated tracks encountering easterlies (westerlies) more likely to move westward (eastward). Under that set up, strong anomalies in steering flow patterns are naturally reflected in the modelled event trajectories, and therefore in the resulting statistics of landfall risk. It is this type of model behaviour that allows translation of regional climate anomalies into shifts in TC landfall risk.






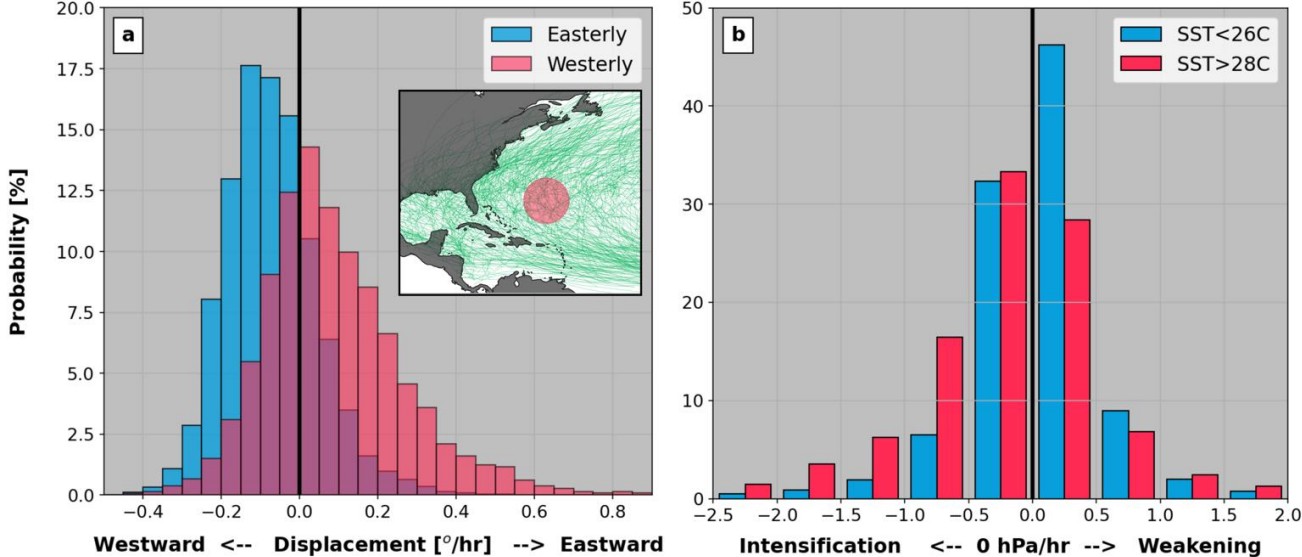

**Fig. 8: (a) Distributions of hourly change in longitude (*dlon*) simulated by the UTC for all events passing through the circle in subpanel (for reference IBTrACS data are displayed in green). Values of *dlon* corresponding to time steps with easterly (westerly) steering flow are shown by the blue (red) distribution. (b) Distributions of hourly change in center pressure (*dCp*) simulated by the UTC for the same region. Values of *dCp* corresponding to time steps with SSTs above 28C are shown in red and those below 26C in blue.**

## 2.2.5 Event intensity

To simulate the intensity evolution over the lifetime of events, two separate algorithms are built. They target:

- the event intensity at genesis point (Center pressure, $Cp_{t=0}$ in mb),
- the increment change in intensity from one step to the next (*dCp* in mb/h)

Both these algorithms are trained in a similar fashion to the *dlat* and *dlon* models. By overlaying ERA5 data onto historical events as reported by IBTRACS we can train quantile regression forest algorithms to approximate conditional distributions (see Appendix A1.4). In both cases, to condition the distribution, we use known storm parameters (*Cp, previous pressure changes, distance to land*) and climate information: wind shear, mean sea level pressure and sea surface temperatures as well as their temporal gradients.

Fig. 8b presents a similar exercise to Fig. 8a where UTC modelled values of *dCp* for all events passing through the same domain. The distributions are split into cases with monthly SSTs above 28 degrees (red) and below 26 degrees (blue). The SST conditioning drives a clear shift toward more likely intensification rate when ocean temperatures reach 28C. As a result, any important anomalies in SSTs are naturally reflected in the modelled UTC event intensities and allow intensification (resp. weakening) to occur over patches of anomalously warm (cold) water. By being closely connected to local environment conditions, the UTC intensity model is able to better capture the evolution of event severity as climate conditions evolve (e.g. see section 3.2).



From a physical point of view, centre pressure is the fundamental measure of storm intensity, and is the logical starting point when modelling event intensification and weakening patterns. In terms of risk measurement however, maximum winds offer a more relevant metric. It is the metric most often reported by media and used to categorize storms in the Saffir Simpson scale; it is also the basis for estimation of TC related damage. As a final step to the intensity module, we therefore translate our *Cp* estimates into maximum wind speeds (1-min sustained winds over water at 10 m, $V_{max}$) following the methods published in

Bruneau et al. (2024).

### 2.2.6 Lysis

Sections 2.2.4 and 2.2.5 describe iterative processes that only terminate once a lysis flag is triggered, typically corresponding to an important weakening of the system. Modelling the cyclone lysis is a difficult exercise due to the small number of data available for training (a single lysis per historical cyclone, most often occurring over ocean). To construct a set of lysis

likelihood targets that goes beyond the binary outcome of historical lysis occurrences, we first assign a probability of lysis to each time step of historical events (see appendix A1.4). A random forest is then trained to predict this probability of lysis from knowledge of event properties, climate conditions as well as the time spent over land and topography set up. When generating events, the random forest algorithm is deployed to predict this probability at each time step. The probability is then used in a binomial draw to sample the survival/lysis outcome.

### 2.3 Model deployment steps

Having described how each of the UTC algorithms is developed we now provide the detailed sequence of steps leading to the generation of UTC event sets:

1. Take monthly gridded data fields for climate state X (e.g. from a historical year of ERA5 reanalysis or alternative gridded climate dataset)

2. Extract $W_{FLD,i}$ weights values for selected $PC_{FLD,i}$ modes of climate state X. Compute associated climate connector values.

3. Model distributions of annual TC numbers by basin conditioned on these connector values for climate state X.

4. Initiate sampling of stochastic years under climate state X.

   For stochastic year 1 to N:

a. Using the modelled distribution from 2.2, sample a number of TC events to simulate for each basin that year.

      b. In each basin, initiate loop over all events.

         For event 1 to nTC:

            i. Sample the genesis point from knowledge of environment conditions in the basin (sea surface

350            temperature and wind shear).





    ii.   Sample the genesis date based on genesis location and environment conditions in the basin

   iii.   Given the genesis location, date and local environmental conditions simulate the distribution of likely starting intensity (*Cp* in mb). Sample intensity value.

   iv.   From knowledge of the above and regional steering conditions, model the distribution of likely latitude and longitude displacements over the following hour. Sample displacements values and move the storm.

    v.   From knowledge of the above, and regional climate conditions, model a distribution for the increment in intensity to expect. Sample increment values and update storm intensity.

   vi.   Model a probability of lysis, and sample lysis occurrence with a binomial draw. If lysis occurs, stop and move to next event, otherwise repeat steps iv, v and vi until lysis occurs.

   vii.   Model 1-min sustained winds over water ($V_{max}$) from *Cp* following Bruneau et al. (2024).





## 3 A probabilistic view of global tropical cyclone risk

In this section we analyse global tropical cyclone risk, as modelled by the UTC under climates of the 1980-2023 period (i.e.
the *deployment period*). The objective is twofold:

UTC evaluation and risk analysis: Ensure that our historical experience over the 1980-2023 period is consistent with the
probabilistic view simulated by the UTC (section 3.1). For that purpose, the UTC is forced with ERA5 reanalysis data for
1980-2023. This dataset provides a view of the climate we have experienced over the post satellite era period where global
TC observations are most reliable.

Counter-factual analysis: Quantify the additional risk variability attributable to uncertainty in the climate experienced over
the period (section 3.2). Here, 1980-2023 climate simulations from the NCAR CESM large ensemble product (LENS2 -
Rodgers et al., 2021) are used to force the UTC. We have selected the 50 smoothed biomass burning (SBMB) ensemble
members to allow sampling of other climate states that could have likely occurred over the 1980 – 2023 period (i.e.
dimension B in section 2.1).

### 3.1 Analysis of global risk patterns and comparison to historical experience

For every year in the 1980-2023 ERA5 reanalysis dataset, we run 2500 samples (i.e. N= 2500 - see step 4 in section 2.3) to
generate 110'000 years of stochastic TC activity globally (44 climates with 2500 realisations of each). Historical observations
over the 1980-2023 period are compared to the UTC probabilistic view of basin wide activity (section 3.1.1), spatial severity
distribution (section 3.1.2) and landfall return period risk (section 3.1.3). It is important to note that historical records should
only be seen as one sample from the distribution of potential risk over the period. The UTC on the other hand is designed to
approximate the full distribution from which the records were sampled. The most important aspect of this evaluation exercise
is therefore to show that the observed sample (i.e. our historical experience) falls within the UTC modelled distribution, with
occurrence statistics that are consistent with the modelled view. There is however no expectation that the observed sample
should fall at the centre of the distribution for all aspects under evaluation.

We choose to analyse and evaluate the modelled 1-min sustained winds ($V_{max}$) as they represent a natural measure of TC
impact. All IBTrACS results are shown using the "USA_WIND" data field from the U.S. Navy Joint Typhoon Warning Center
(JTWC) as it is based on a globally consistent and well documented methodology (Knapp and Kruk 2010).

### 3.1.1 Global TC activity

Fig. 9 shows the density distribution of annual named storm numbers in the 110'000-year stochastic event set for each of the
active basins, and for each TC season of ERA5 forcing (1980-2023). They represent the likelihood of outcomes under each of
the historical annual climate states (1-99[th] percentile intervals are displayed in Figure 9). Observed records are overlayed on



all the distributions as white circles. They represent the one outcome that occurred under that observed climate state. As such we should not expect to see the white circles at the centre of the UTC distributions for all individual years, however it is important that the circles do fall within the modelled distributions and that the overall statistics are in line with the UTC data. Over the full Fig. 9 dataset, observed occurrence levels are within the UTC 50[th] confidence interval in 64% of cases while 94% fall within the 90% confidence.

In all basins the season-to-season variability in the modelled UTC distributions is consistent with variability in observed TC numbers. This ability to capture season -to- season variability is only possible thanks to the climate-connected nature of the model. Without climate conditioning, every year would be assigned the same (static) activity distribution by basin (the average distribution over the period, see right panels in Fig. 9). This has important consequences when trying to assess short term risk variability (e.g. the seasonal trends) as well as global connections in TC activity. For instance, by grouping the data from Fig. 9 in terms of climate regimes known to impact TC activity we can quantify the shifts in basin-wide TC activity attributable to physical cycles such as ENSO, and assess how activity levels in different basins are connected (e.g. anticorrelation between North Atlantic and East Pacific, Steptoe et al. 2017). Both these aspects will be explored in separate studies where we illustrate the value of the UTC as a tool for seasonal risk forecasting, as well as for quantification of global risk correlations.





Fig. 9: Distributions of named storms numbers from a 110'000-year UTC dataset forced by ERA5 reanalysis data of the 1980-2023 period for (a) North Atlantic, (b) East Pacific, (c) western North Pacific, (d) North Indian, (e) South Pacific and (f) South Indian basins. Historical occurrences from the Colorado State University database are shown as white dots and UTC distributions averaged over the whole period (climatology) are shown in the right panels. The simulated 1-99[th], 10-90[th], 20-80[th], 30-70[th] and 40-60[th] intervals are displayed.



### 3.1.2 TC spatial risk distribution across the globe

Beyond basin-wide activity numbers, one of the main goals of a TC stochastic event set is to capture spatial variability in risk severity within each basin. Observation records over the 44-year period of post-satellite-era are too scarce to provide a complete view of risk, but they do highlight regions where TC risk is concentrated. Analysis of maximum sustained TC winds globally over the 1980-2023 period (Fig. 10a) shows the eastern Philippines as the riskiest region on earth, followed by the western Caribbean. A closer look at the Gulf of Mexico or Florida regions reveals discontinuous patterns where important historical events are clearly identifiable among lower risk neighbouring levels. Such discontinuities in risk mapping are due to an insufficient number of seasons in the historical records to fully capture the spatial risk distribution of extreme events (the risk distribution is under-sampled).

Using 110'000 years of simulations from the UTC, we can assemble a more consistent view of risk (Fig. 10b). The 110'000-year UTC event set is here split in 2500 groups of 44 years (i.e. 2500 iterations of the 1980-2023 period) allowing computation of 2500 equivalent versions of Fig. 10a. Fig 10b represents the grid cell average (2.5-degree resolution) of these 2500 versions and captures the *expected peak winds* over the 1980-2023 period for each grid cell. In other words, given what the UTC has learnt from global historical records, and the role played by climate physics in driving TC risk, peak winds of the magnitude reported in Fig. 10b should be expected when experiencing the climate of the 1980-2023 period. To gain some insight into the main drivers influencing the UTC view of risk, we also provide maps of peak season climatology (i.e. August – October in the Northern hemisphere and December - February in Southern hemisphere) for SST, SHR, U850 and V850 fields (Fig. 11).

For the North Atlantic (NA) basin the regions of peak risk are very consistent with historical records. This is an important result as the NA is the region of the world where we have access to the best quality of observation records over the 1980-2023 period. Category 5 level winds (i.e. 1 min sustained winds of 70 m/s – the white contours in Fig. 10) are expected to occur during the period for regions along the Caribbean islands, Gulf of Mexico as well as southern Florida (Fig. 10b). This is in line with historical records (Fig. 10a) and directly relatable to favourable peak season climatological conditions with warm SSTs (above 28C, Fig. 11a) and weak vertical wind shear (below 10 m/s, Fig. 11b). Patterns of cooling SSTs, higher vertical wind shear as well as a strong westerly component of the steering flow (Fig. 11c) also clearly help understand the reduction in risk north of the Florida coast.

For the Eastern Pacific (EP) basin, the regions of Category 5 level expected winds are again well aligned with historical evidence (Fig. 10) and coincide with favourable environmental conditions (Fig. 11) on the eastern side of the basin. Further west, a notable patch of large vertical wind shear is present over the Hawaiian Islands (Fig. 11b), along with a northerly component to the steering flow (Fig. 11c) that tend to protect the islands. and translate into reduced risk levels both in terms of UTC expectations (Fig. 10b) and historical experience (Fig. 10a). More favourable conditions to the south of the islands allow for increased risk levels.

The western North Pacific (WNP) basin shows a good level of agreement between historical experience and UTC expectations. Environmental conditions in the basin are mostly favourable up to the Japanese coast with very warm SSTs (Fig. 11a) and





weak vertical shear (Fig. 11b). This translates into a wide region of peak risk to the east and north of the Philippines. As is the case in the eastern coast of the US, there is an important decrease in risk when moving north towards central Japan, with a

sharp SST gradient, strong vertical wind shear and a dominant westerly steering flow.

TC activity patterns in the North Indian basin are consistent between model expectations and historical experience (Fig.10), with higher risk localized in the north of the Bay of Bengal. The presence of high vertical wind shear to the south acts to dampen activity, however we note that peak activity in the basin does not occur during the August-October period displayed in Fig. 11 and as a result modelled patterns are not further analysed from a physical point of view.

Largest discrepancies between UTC expectations and observation records occur in the Southern hemisphere (SH). In particular the UTC has wider expectations of peak winds for the North East (east of Townsville) and North West of Australia (north of Port Hedland) as well as for most of the northern part of the South Indian (SI) ocean (Fig. 10b). Environmental conditions during peak SH season are mostly favourable in these three regions (Fig. 11) which helps understand why the UTC expected levels are high. With SSTs around the 28C level and wind shear conditions below the 10 m/s threshold, these areas are

comparable to the Caribbean and southern Gulf of Mexico regions. Consequently, UTC expectations in terms of peak winds are of similar magnitudes (i.e. at the category 5 level). While it is possible that the discrepancies could be attributable to a model bias (e.g. missing some local physics) or over generalization, it is also likely that events in these regions are under reported in IBTrACS. Coverage of TC events in the South Indian in particular, is likely not as thorough as in other basins. In the north East and North West of Australia, historical records are in line with the UTC expectations at the coast (see also Fig.

13), but differ further out at sea. Here again it is possible that events have been underreported away from their direct landfall impacts (no immediate risk to population). The alternative would be that events tend to only reach their peaks at the coast, and the physical reason for such behaviour is missed by our modelling approach / resolution. Steering flow patterns (Fig 11c,d) around Port Hedland are worth noting as they help explain the concentration of risk observed historically over that section of the coast (see Figure 13r). With a corridor of strong northerly steering above warm SSTs and a weak shear environment leading

to increased risk expectations.



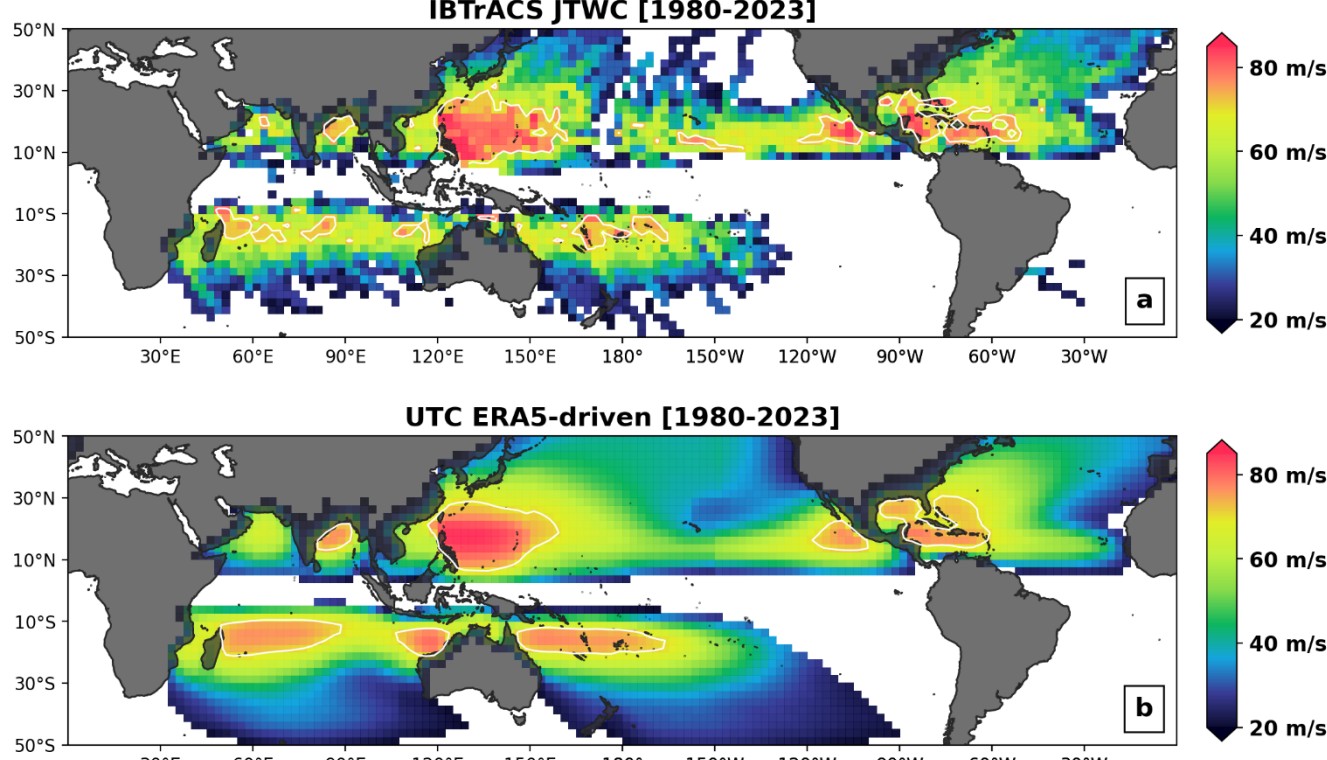

**Fig. 10: 2.5-degree resolution map of (a) peak 1-min sustained winds recorded in IBTRACS over the 1980-2023 period and (b)** 475 **expected 44-year peak sustained winds from the ERA5-driven UTC stochastic set (110'000 years of activity). The white contours represent the 70 m/s wind level (Cat5 threshold).**








**Fig. 11: peak season climatology (Aug-Oct in Northern hemisphere and Dec – Feb in Southern hemisphere for 1980-2023) for ocean surface temperatures (a), vertical wind shear (b) as well as meridional (c) and zonal (d) components of the 850hpa winds. Contour lines correspond to 28C ocean temperatures (in panel a), 10m/s wind shear (panel b) and the 0m/s steering (panels c and d).**




The analysis of Fig. 10 and 11 shows that the UTC, driven by physical patterns known to be important to TC risk, is able to reproduce a global TC risk severity distribution that is consistent with historical records. As such it provides a reliable tool to assess the impact of climate phases linked to shifts in patterns of ocean temperatures, vertical wind shear or steering currents. As an example, we here briefly discuss the influence of the ENSO cycle. La Nina events are characterized by a western shift

in warm ocean temperatures in the Pacific, with a resulting change in atmospheric circulation leading to reduced (increased) vertical wind shear in the Gulf of Mexico and Caribbean Sea regions (in the eastern Pacific). With these shifts in climate patterns directly conditioning the UTC event generation algorithms, the framework can be used to quantify the impact of the cycle on global TC risk. Here we show the annual occurrence of Category 3 events across the globe as modelled by the UTC for El Niño (Fig 12a) and La Niña (Fig. 12b) conditions. The differences between both states is also shown in Fig. 12c.

Consistent with the shifts in climate conditions described above, the risk of major hurricane occurrence in the Gulf of Mexico and Caribbean Sea under La Niña conditions is over 50% higher than during El Niño conditions with the opposite happening along the western Mexican coast. In the western Pacific, during El Niño conditions, the occurrence risk for category 3 events and above is increasing by up to 50% on the eastern side of the basin, while the risk around the Philippines and the Chinese coast is decreasing. In the Southern Hemisphere, during La Niña conditions, Category 3 occurrence risk increases around the

Australian coast and decreases for the Pacific Islands.







Fig. 12: Annual occurrence rates of category 3 tropical cyclones for 2 subsets of the 110'000-year UTC event set, characterizing the 10 strongest El Niño (a) and La Niña (b) seasons in the period 1980-2023; each hemisphere uses the ENSO conditions of the in-season months (ASO for the Northern hemisphere and DJF for the Southern hemisphere). The absolute differences between El Niño and La Niña rates of Cat3+ are shown in the last panel (c).



### 3.1.3 TC landfall risk statistics

From the perspective of risk to society, the most relevant aspect to analyse is the severity distribution at landfall. We here focus on major metropolitan coastal region across the globe and use the 110'000 years of UTC activity to assess return period wind speed levels at landfall (Fig. 13). For each region, the intensity of events is recorded for both the historical dataset and

the UTC stochastic set. By ranking events in terms of their peak intensity in a 100 km circle centred on a given city (y-axis, Fig. 13) we can then compute the associated return periods (x-axis, Fig. 13).

For all panels, the 44 years of observations are reported as black dots, with the most intense events for each region located at the 44 years return period level. Quantifying the severity of rare extreme events from such a short record of observation years is an obvious issue, and here again the UTC offers an alternative by providing 110'000 years of activity. Return period intensity

levels from the UTC are presented in solid-coloured lines up to return periods of 1 in 300 years (i.e. beyond any available reliable historical records). Uncertainty bands are also reported by overlaying the regions covered by UTC subsets of 44 years (shaded areas). The darker shaded region captures the spread between the 5th and 95th percentiles from all 44 years subsets. The lighter shaded region extends to the entire dataset showing the spread between the two most extreme 44 years subsets available in the 110'000 years simulation.

In all cases the historical records fall within the range covered by the lighter shaded area, showing that our historical experience is contained within the distribution modelled by the UTC. The majority of data points are also contained in the range covered by the 5th to 95th quantiles, with the relative positions of the modelled averaged view (solid line) and historical records (dark line with dots) varying from one city to another. The modelled view is sometimes suggesting higher (e.g. Hong Kong – Fig. 13o) or lower (e.g. New York – Fig. 13d) expected risk than experienced. As was the case for the discussion of Fig. 11, these

discrepancies are mostly attributable to the limited length of historical records (under sampling of the risk) but could also be the result of local physical patterns being missed by the UTC modelling framework. The spatial resolution of the forcing climate data can for instance be too coarse to capture local steering shifts or sharp SST gradients in some regions (for example around the Gulf Stream).





Fig. 13: Return period of maximum 1-min sustained wind levels (m/s) in major coastal cities across the 6 active TC basins of the world. The thick dark-dotted line provides historical information for the period 1980-2023. The light shaded regions show the whole range simulated by the model considering 2500 samples of 44 years while the stronger shaded regions show the 5-95$^{th}$ interval. Finally, the coloured line illustrates the converged view, aggregating the 110'000 years of stochastic simulations.



## 3.2 Impact of climate variability.

The exercise presented in section 3.1 focused on analysing TC risk under the climate conditions observed between 1980 and 2023. Here we expand the analysis to consider other climate conditions that could have occurred over the period (i.e dimension B in section 2.1). For this purpose, we ran an additional 550'000 years of stochastic TC activity based on the 50 smoothed biomass burning members of the CESM LENS2 (Rodgers et al., 2021) climate outputs for 1980-2023 (i.e. N = 250 samples, for 50 members, each covering the 44 years period). This allows reproduction of the analysis in Fig. 13, accounting for

variability in the climate of the 1980-2023 period.

Fig. 14 below shows 60 curves capturing the return period risk for events crossing segments of the coast in Louisiana (Fig. 14a) and the Carolinas (Fig. 14b). The 50 blue lines correspond to the 50 CESM LENS2 members and are each built from 11'000 years of stochastic activity, they represent 50 different views of the climate over the 1980-2023 period. The 10 red lines are subsets of the ERA5 forced stochastic set (i.e. the one analysed in section 3a). They are also built from 11'000 years

of activity, but are all representative of the exact same climate over the 1980-2023 period (the one we experienced). By including additional climate conditions, the CESM driven curves (blue) cover a wider spread than when only the ERA5 climate is considered (red curves).

This missing variability is important: while the narrow spread of the red curves provides the impression that the modelled view of risk has converged, it is important to acknowledge that it is only sampling dimension A. Therefore, it has converged under

the assumption that the only climate we could have observed over the 1980-2023 period is the one portrayed by ERA5. When considering the alternative climates from CESM, the spread widens (blue curves). The UTC is now sampling a more complete risk distribution.

At the Category 5 wind speed threshold of 70 m/s for instance the red curves all assign a return period between 40-45 years in Louisiana (Fig. 14a). The range covered by the blue curves on the other hand goes from 40 to 80 years, suggesting that return

period for Cat5 winds along that section of the coast could be much lower and that the climate experienced along the Louisiana coast may have been on the unlucky side with regards to its influence on hurricane risk. At the same threshold of 70 m/s along the Carolina coast, the ERA5 driven view of risk is once again converged towards an estimate of 160-180 years. However, once we allow sampling from the wider range of climates covered by the CESM simulations that estimates ranges from 100 to 300 years. Note similar patterns can be observed at other intensity thresholds, with the width of the CESM spread narrowing

for lower winds.

The ability to sample dimension B comes with a heavy computing cost (5x dimension A in the example above). Yet, the analysis of Fig. 14 shows it has an important impact on our estimates of landfall risk distributions. The inclusion of that extra layer of risk variability is even more important when considering risk under future climate conditions, given the larger associated uncertainty. The use of the UTC to derive forward looking TC event set will be presented in a follow up study, with

two key use cases:



1. Deploy the UTC with seasonal climate projections (e.g. 51 members from the ECMWF monthly seasonal forecast): this allows generation of a seasonally conditioned TC stochastic event set, translating projected anomalies in SST, wind steering and vertical wind shear patterns into regional shifts in TC risk for the season ahead.

2. Deploy the UTC with future climate projections (e.g. CESM members for the 2025-2100 period). This allows analysis of projected shifts in regional TC risk over the coming decades, under varying levels of global warming.

In both use cases above, the inclusion of dimension B is critical. Focussing only on one climate path vastly under-samples the

full risk distribution, with important implications in terms of risk management and mitigation.

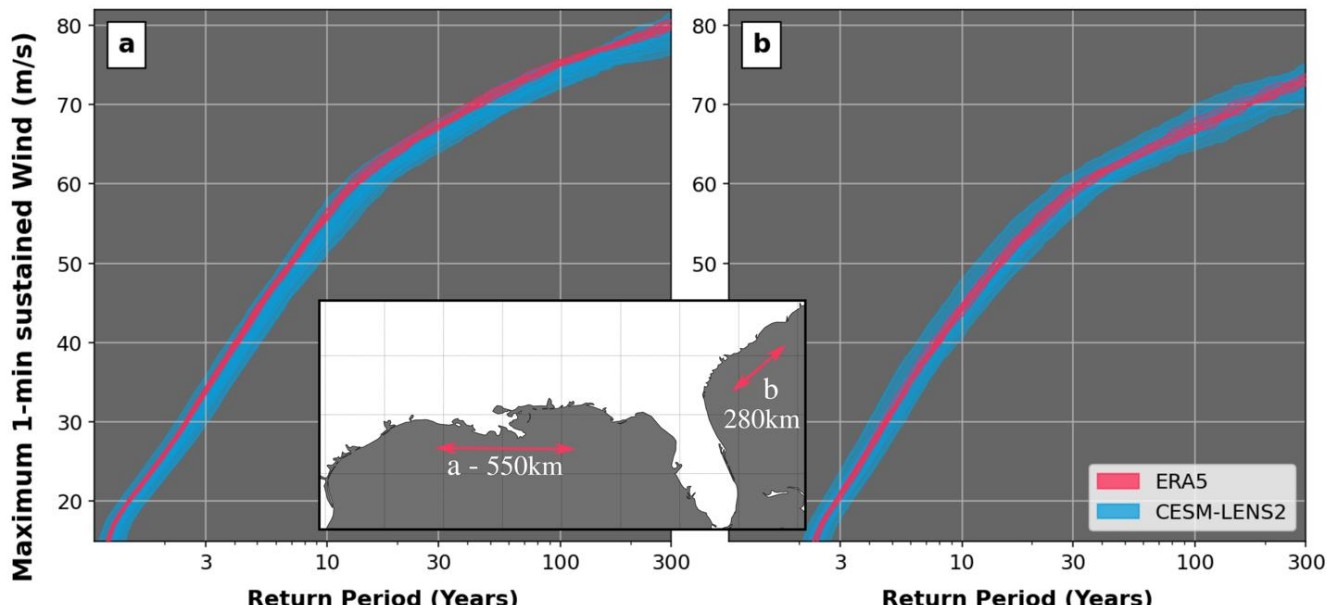

**Fig. 14: Return period of maximum 1-min sustained wind levels (m/s) along two segments of the US coast (inset) for 10 sets of 11'000 years event set driven by ERA5 1980-2023 climate (red curves) and 50 sets of 11'000 years driven by the CESM-ENS2 over the same period (blue curves).**






## 4. Conclusion

Stochastic event sets have helped understand and manage Tropical Cyclone (TC) risk for over three decades. They are particularly useful in quantifying tail risk far beyond historical experience. However, to date they have mostly been built from statistical relationships fit to historical records, without accounting for variability in the state of the climate. We here present an alternative approach (Unified Tropical Cyclone – UTC) where we explicitly connect the event generation algorithms to an input climate state. After an initial description of all algorithms involved in the climate-connected event generation process, we show results from two stochastic event sets representative of the 1980-2023 climate.

First, we force the UTC with reanalysis data over the period and compare the resulting view of risk to historical records. This analysis shows that the UTC modelled view is consistent with our historical experience over the past 44 years. It also highlights known limitations when it comes to assessing risk levels from historical records only: the period of reliable global records is too small and as a result (i) it under samples the spatial distribution of risk (Fig. 10) and (ii) misrepresents the likelihood of rare extreme events (i.e. tail risk, Fig. 12). Using 110'000 years of stochastic activity from the UTC helps analyse global TC risk beyond these limitations.

We then extend the analysis using additional forcing from a global climate model (CESM LENS2) over the same period to quantify the impact of climate variability. Accounting for this additional dimension of risk variability increases the sampling space (Fig. 14) and allows analysis of physically realistic scenarios that fall outside of the scope covered by traditional TC risk assessment frameworks.

The natural next step is to deploy the UTC under forward looking climate scenarios, allowing risk assessment in the context of the season ahead, or mitigation and planning strategies for the decades ahead. For such applications, the need to include sampling of climate variability (dimension B) is even more important, given the additional uncertainty associated with future climate projections and pathways.



# APPENDIX

**A1: Technical description of event generation algorithms**

**A1.1. Principal component analysis of gridded climate data**

In machine learning terminology, Principal Component Analysis (PCA) falls into the category of unsupervised learning algorithms. These are methods designed to identify patterns in large datasets, without being told what to look for (the learning step does not involve any explicit target). Thanks to PCA, gridded fields can be decomposed into a series of patterns ranked

in terms of how much data variability they explain. A typical use of PCA is then to select only a subset of all patterns (those that explain most of the variability) to reduce the dimension of the original dataset. The approach applied here differs slightly in that the subset of patterns are selected based on how they correlate to TC activity in various basins of the world (e.g. see Fig. 4).

Prior to PCA, the raw gridded fields are standardized via a centering / scaling step, where the time average of each cell is

removed (centering) before normalization by the cell standard deviation over the time dimension (scaling). For each climate field (i.e. FLD = SST, MSLP, U850, U250, V850, V250 or SHR - see section 2.2) a PCA is then performed on the standardized array. This results in the projection of the gridded fields into a set of orthogonal vectors (Principal Components – $PC_{FLD,i}$ - see Fig. 3) with associated coordinates, or weights ($W_{FLD,i}$).

For a given field (e.g. FLD = SST) the full decomposition of the raw monthly arrays can be formulated as follows:

$$FLD(t) = MU_{FLD} + SD_{FLD} * \sum_{i=1}^{N_{pc}} W_{FLD,i}(t) PC_{FLD,i} \qquad (A.1)$$

Where $MU_{FLD}$ is the time averaged field array, $SD_{FLD}$ the standard deviation array, $N_{pc}$ is the total number of principal components and t is the time variable (monthly resolution). Note that the decomposition above can also be applied to stacked combinations of fields where several arrays are layered. These will be referred to with a "+" sign in what follows (e.g. FLD = SST+SHR+MSLP, see connector 3 in A2).


**A1.2. List of climate connectors selected to condition TC activity**

By computing the correlation between TC activity in each of the world's active basin and the $W_{FLD,i}$ averaged over several months covering peak season activity, we can identify $PC_{FLD,i}$ fields that are good candidates to connect climate state and TC occurrence (see Fig. 4). After screening these candidates to ensure the physical reasons for the correlation are understood (see

example in section 2.2.1) we end up with a selection of 13 $PC_{FLD,i}$ physical patterns to characterize a global climate state. The scalar values below are referred to as *climate connectors* and are used to condition TC activity globally:



**Table A.1: list of connectors selected to condition the UTC model.**

| Basin | Region | Variable | PCA | Period | Formulation |
|---|---|---|---|---|---|
| Northern Hemisphere (NA, WP & EP) | Global | SST | PCA 3 | July-November | $C_{NA,1}(k) = \frac{1}{5} \sum_{t=July(k)}^{t=November(k)} W_{SST,3}(t)$<br>Where k is an index for the year. |
| North Atlantic (NA) | NA only | MSLP | PCA 3 | July-September | $C_{NA,2}(k) = \frac{1}{3} \sum_{t=August(k)}^{t=October(k)} W_{MSLP,3}(t)$ |
| North Atlantic | NA only | SST, SHR, MSLP | PCA 1 | July-November | $C_{NA,3}(k) = \frac{1}{3} \sum_{t=July(k)}^{t=September(k)} W_{SST+SHR+MSLP,1}(t)$ |
| Western North Pacific (WP) | WP only | Usteering, SHR | PCA 6 | April-August | idem |
| Eastern North Pacific (EP) | EP only | SST, SHR, MSLP | PCA 3 | September-November | … |
| North Indian (NI) | NI only | SHR | PCA 9 | April-August | … |
| North Indian (NI) | NI only | Usteering, SST | PCA 6 | March-September | … |
| Southern Hemisphere | Global | SST | PCA 3 | January-March | … |
| South Western Indian | Global | SHR | PCA 8 | February-April | … |
| Australia | Global | Usteering, SST, MSLP | PCA 5 | December-February | … |
| Australia | Australia | SHR, MSLP | PCA 9 | August-September | … |
| Southern Pacific | Pacific | Usteering, SHR, MSLP | PCA 4 | February-April | … |
| Southern Pacific | Pacific | Usteering, SHR, MSLP | PCA 8 | January-March | … |






### A1.3. Bayesian Generalized Linear Model for TC annual frequency

Bayesian Generalized Linear Models (GLMs) are a common approach to model conditional distributions when the training data is scarce (Elsner and Jagger, 2004). In this study we use a Poisson GLM to model the conditional distribution of annual
tropical cyclones in each basin. The λ rate from the Poisson distribution is conditioned on selected climate connectors (see A1.2). We use TC storm count data by season for the 1980-2020 period to train the relationship between λ and the climate state as described by the selected connector values.

As an example, the simplified model of Fig. 5 is presented below:

$$\text{nTC}_k \sim \text{Poisson}(\lambda_k)$$
$$\log(\lambda_k) = \beta_0 + \beta_1 C_{NA,1}(k) \tag{A.2}$$

Where k is an index for the season of interest, $\text{nTC}_k$ represents the number of storms occurring in that year (season). A Poisson process is assumed with parameter $\lambda_k$ referring to the rate for season k and the logarithm of $\lambda_k$ is modelled as a function of the July – November average $W_{SST,3}$ (i.e. connector 1 in A1.2 and the relationship shown in Figure 4). The parameter vector β = ( $\beta_0$, $\beta_1$) is specified by a multivariate normal distribution as discussed in Elsner and Jagger (2004).


The complete model, as implemented in the UTC involves 3 connectors for the North Atlantic basin:

$$\log(\lambda_k) = \beta_0 + \beta_1 C_{NA,1}(k) + \beta_2 C_{NA,2}(k) + \beta_3 C_{NA,3}(k) \tag{A.3}$$

Genesis models for all other active basins follow a similar structure, with the description of connectors involved in each model detailed in Table A.1.

**A1.4: Random forests and quantile regression forest algorithms**

Supervised learning refers to the sub-group of ML algorithms that require an explicit target during the training phase. When trained with a large volume of data, and carefully set up to avoid overfitting issues, these algorithms offer a very powerful tool to extract relationships without the need for a human to parametrize and tune their form. In this study we rely mainly on a family of such algorithms called random forests, because of their flexibility and robustness with regard to the overfitting issue.
The building block at the core of the random forest algorithm is the decision tree. Decision trees are essentially a series of nested if-then statements designed to recursively partition the training data into smaller groups that are more homogeneous with regards to the target. They are very popular due to their transparency and ease of interpretation. However, they are also known to lack stability and small perturbations in the data can lead to very different tree architectures.

The random forest algorithm (Breiman 2001) was developed to overcome these limitations, by grouping together a large
number of decision trees trained under slightly different conditions (random subsets of the input data and features at each tree





nodes). The resulting ensemble of trees is then used as a cohort and the prediction from the forest is obtained by averaging each tree's vote. This greatly reduces challenges of overfitting and leads to a much more stable algorithm.

When used on new data to make a prediction, a random forest estimates the mean from the outcome distribution. In some cases, such as the example of section 2.2.6, this is sufficient information. However, in other cases it is desirable to retain
information about the entire outcome distribution rather than simply focus on the mean prediction.

For such cases we leverage an algorithm called quantile regression forest (Meinshausen, 2006), designed to keep the value of all observations in the terminal leaves to allow assessment of the full conditional distribution when making prediction on new data. This contrasts with the standard random forest where only the mean of observations in the leaves is kept (see Loridan et al. 2017 for more details).


In section 2.2.6 we use a random forest algorithm to model lysis probability. To train the algorithm we first assign a target lysis probability value to all historical TC track points in the IBTRACS database. Note the target probabilities are capped at 0.5, acknowledging that some ambiguity can exist around the decision to stop reporting an event (i.e. the last point is not an exact representation of the time of lysis):

- Points for which a lysis occurred (last point recorded for a given event) are assigned a value of 0.5.
- Points within 24h of lysis are assigned a value between 0 and 0.5 to reflect the belief that lysis was a likely possibility, following a simple linear law: $P_{lysis}(t_{lysis} - t) = \frac{1}{2t}$
- All others points are assigned a value of 0.

We then train a random forest to predict this scalar value. Pressure field and pressure change to previous time step, climate conditions (MSL, SST, SHR and their spatial gradient at the storm location), topography, time spent overland and distance travelled overland are provided to the random forest algorithm to predict the probability of lysis.

Quantile regression forest (qrf) algorithms are the building block of our Markov Chain Monte Carlo (MCMC) modelling of
event trajectories and intensity evolution. We train the various qrf involved to predict distributions of the hourly changes in latitude, longitude and center pressure from knowledge of the event parameters at current and previous step as well as the local environment (SST, vertical wind shear and steering flow). The end result is a collection of qrf algorithms able to efficiently generate conditional distributions on the fly at every event time step knowing the state of the climate, therefore allowing sampling of all parameters needed to update the track to its next state (i.e. next center position and intensity).




**Author contribution:**

Loridan and Bruneau jointly designed the algorithms described in this study. Bruneau developed most of the model code and performed the simulations with partial contributions from Loridan. Loridan prepared the manuscript with contributions from Bruneau.

**Competing interests**

The authors declare that they have no conflict of interest.

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
