# Peer review of "Reask UTC: a machine learning modeling framework to generate climate connected tropical cyclone event sets globally."

_EGUsphere, 2024_

## Author Response (AR1)

**Author's reply' to the reviewers**

**RC1**: 'Comment on egusphere-2024-3253', Ralf Toumi, 07 Nov 2024

*A very good paper with many interesting results. I only have minor comments largely focussed on giving more quantification of performance:*

We thank the reviewer for a thorough and very timely review.

Thanks to comment #2 we realized that our description of the genesis module was incomplete. We have now added a paragraph (l220-235) to the main text as well as extra information in the appendix to clarify how the genesis numbers are distributed from basin-wide initial estimates to regional sub-domains.

*Fig. 4. Give R^2 and p-value*

Values below were added to the caption for Fig. 4

R :-0.66

$R^2$: 0.44

P-value: 1.11e-06

*Fig 6. and 7. Showing/selecting individual years is only illustrative. The authors should give a metric of overall performance.*

As acknowledged in the manuscript (l239-242), our proposed genesis likelihood parametrization is very simple. It is designed to ensure climate variables known to influence TC formation are used to "nudge" static genesis probabilities in a direction consistent with a favourable/unfavourable environment set up.

Our starting point is to estimate the probabilities based on a static smoothing of historical occurrence rates ($P_{static}$ – Fig. 6a). This is a method commonly applied when building TC stochastic event sets, yet it fails to capture any direct influence of climate variability. We therefore add a simple physics-based scaling to allow SST and shear variables to dynamically refine the probabilities ($P_{dyn}$).

To assess performance beyond our initial qualitative analysis we have now performed a simple evaluation exercise (now added to the main text – l270-279): For all 1980-2020 observed genesis occurrence, we compute the genesis probability under both the static model ($P_{static}$, Fig.6a) and with the dynamic scaling of eq. 1 ($P_{dyn}$).

Over the full dataset of all historical occurrence the average climate conditioned genesis probability $P_{dyn}$ is increased by up to 14.8% compared to the static version ($P_{static}$): i.e. from

additional knowledge about the environmental set up, the dynamical model is statistically increasing genesis likelihood in regions where genesis has occurred. The expected number of TC genesis is higher under the $P_{dyn}$ model than it is under $P_{static}$

Given the scarcity of observed positive genesis outcomes many traditional performance metrics offer limited insight. We believe the analysis above shows that adding a climate conditioning to a static estimate helps improve likelihood estimates while also ensuring the model reacts to important physical changes in the climate.

Fig. 9. Give RMSE of predicted mean count vs observed.

The table below was added to the main text

Table: Statistical analysis of the yearly storm activity for each basin (see Figure 9).

|  | **NA** | **WP** | **EP** | **NI** | **SI** | **SP** |
|---|---|---|---|---|---|---|
| Climatology obs (modelled) | 13.5 (13.5) | 25.3 (25.5) | 17.3 (17.5) | 5.07 (4.95) | 15.65 (15) | 9.4 (10.3) |
| RMSE (#) | 3.17 | 4.30 | 3.78 | 1.48 | 2.55 | 2.86 |
| BIAS (#) | 0.01 | 0.19 | 0.19 | -0.12 | -0.65 | 0.9 |

Regarding both the scaling and the genesis probability it would be useful to understand how relatively important the individual terms are. Genesis indices are notoriously poor at describing the interannual variation and it seems the authors are presenting a new genesis index.  I would be particularly interested in the monthly shear. The synoptic shear is important, but it is not obvious to me that the monthly mean anomalies capture the synoptic variability.

To help address this question we have performed a similar exercise as for comment #2. For all historical genesis occurrence, we have extracted the different parameters that form the scaling factors of equation 1. Given that we know genesis occurred, we expect the contributions of each parameter in the figure below to be (more often than not) towards increasing the genesis probability estimates (i.e. > 1). The combined SST/SHR factor is > 1 in 59% of cases (right panel), showing, as was the case in comment #2, that the method does nudge the probability in the correct direction. Out of the two factors (SST and 1/SHR) it is the shear component that is actually bringing more value (> 1 in 59% of cases compared to 55% for SST).

As discussed above, our intentions with this simple physics-based parametrization is to avoid using a static genesis model and enrich it with climate variables that are known to favour TC formation. We acknowledge this is perhaps the weakest part the UTC modelling system and aim to improve the component in the future.

[Figure]

l.294 Perhaps a reference to the crucial role of steering for landfall risk: https://rmets.onlinelibrary.wiley.com/doi/10.1002/asl.101

This does not seem to be the right reference?

**Sec 3.2. No mention of CESM biases, so the added value of the longer runs may be ambiguous?**

This was indeed an omission from our side, and a description of our approach to bias correction has now been added (l591-600 – see also below).

Unlike the ERA5 dataset, the CESM-LENS2 is not a reanalysis product. The data used in this study come from climate model runs without extensive assimilation of historical observations. While this is important to allow sampling of climate variability over the period, it also provides well documented challenges in terms of model biases (Simpson et al. 2020, Lee et al. 2021).

To address this issue, we first compute a pixel-by-pixel and per-month climatology across all the members for each CESM climate variable from section 2.1. We also compute and store a bias correction that aligns the data to the 1980-2020 ERA5 dataset. This bias correction is then applied to each member separately, ensuring that:

- on average, all the members are unbiased to ERA5 over the period 1980-2020,
- The relative variability of each member is maintained after the correction is applied

The stochastic event set used in this section is generated on this bias-corrected version of CESM-LENS2, providing alternative climate TC risk consistent with what has been observed (ERA5) but through different paths.

**All Equations should be numbered**

done

RC2: 'Comment on egusphere-2024-3253',

Congratulations on building such a model and having it perform so well! I enjoyed reading your manuscript, it reads very well and I only have a couple of very minor comments, see below.

-- Reviewed by Dr. Nadia Bloemendaal (and, as I need to disclose whether someone helped me: my 2-year old daughter who has been sitting and coloring beside me ;-))

Thank you so much, we appreciate the kind words and the thorough review. Also, congratulations on training the next generation of climate scientists from such a young age.

- Please add numbers to the equations.

  done

- Line 156: Which intensity metric did you take from IBTrACS – and if you took Vmax: did you also correct for the fact that different ocean basins in IBTrACS have different wind recording standards (in case you took the WMO_WIND variable)? See IBTrACS documentation for more information on this: https://www.ncei.noaa.gov/sites/default/files/2021-07/IBTrACS_v04_column_documentation.pdf

  Yes, we take Vmax. We use the USA agencies 1-min sustained winds (Vmax) as these are consistently available globally (1-min sustained for all basins).

  This is clarified on l 415-416 as follows: "All IBTrACS results are shown using the "USA_WIND" data field from HURDAT database (Landsea and Frankin, 2013) and the U.S. Navy Joint Typhoon Warning Center (JTWC) as it is based on a globally consistent and well documented methodology (Knapp and Kruk 2010)."

- It is not clear why the equation on line 261 is the way it is – what determined to square the probability of the wind shear?

  This question is similar to the discussion with Reviewer 1 on comments #2 and #4, where we clarify our intentions when formulating these simple physics-based parametrizations. We start from a static baseline, where historical records are used to define default estimates. This is the standard approach in TC stochastic event generation, but it fails to account for climate variability. As a first step towards a more dynamic approach, we then add scaling parameters based on climate variables known to impact TC formation. This is similar in concepts to other genesis potential indices, but we keep the level of tuning of our formulation to a minimum.

  The decision to square the wind shear component in eq. 2 was motivated by a desire to increase the importance of the term relative to SST and MSLP (see also answer to comment #4 by reviewer 1) as in our early development phase, it was improving the

narrow temporal window of activity in the Main Development Region particularly (mostly only active over 2 months). However, in this final version weighted with the historical information, there is little impact in squaring the wind shear or not.

- Line 270: I think you misspelled my last name, it's Bloemendaal (and not the German equivalent 😵)

Done (apologies!)

- Line 324: Can you briefly explain these methods?

*In Bruneau et al. (2024), $V_m$* is modeled indirectly through the estimation of the coefficient $\alpha$ in $V_m = \alpha \Delta p^{1/2}$. This $\alpha$ parameter can be related to the Holland *B* parameter as $B = \rho e \alpha^2$ (Knaff et al. 2011b). The power 1/2 in the equation represents cyclostrophic balance (Knaff and Zehr 2007).

The $\alpha$ parameter is modelled via quantile regression forest, and the features consist of centre pressure at t and t-1, the mean sea level pressure and wind shear as well as a flag for the basin and the knowledge of being overwater or overland.

 A summary of the above has been added to the manuscript (l350-352)

Line 372 – 376: Can you comment on how well this ensemble product performs in simulating the climate? Is there an ENSO bias present in this model that could potentially affect your outcomes? (i.e. see Seager et al., 2019 https://www.nature.com/articles/s41558-019-0505-x )

Chen et al. 2021 (chen_etal_jc2021.pdf) have investigated ENSO in E3SM-1-0, CESM2, and GFDL-CM4 models, and they show that capturing the full ENSO dynamic is still challenging but they highlight that models are getting better in capturing ENSO characteristics than their predecessors (including amplitude, time scale, spatial patterns).

We have now added a paragraph discussing the measures we took to bias-correct the CESM data (l591-600). We do however acknowledge that by using a single climate model (even with a large ensemble of members) we are exposed to potential model biases (including the representation of the ENSO cycle – see Chen et al. reference above). To address this issue we would need to run a similar experiment with alternative sources of climate data, which is beyond the scope of the current study (but something we are planning to do in the future).

A comment was added in the conclusion (l659-661).

More fundamentally, and as pointed out by the reviewer via the article referenced, there is also a chance that this ENSO bias is more fundamental to all current generation climate models. Here we do not offer a ready solution but still believe our

results to be of value in understanding how TC risk could differ in physically realistic alternative climates.

- Could you also model natural variability like the MJO?

The forcing data we use is monthly, which makes it difficult to simulate intra-seasonal modes of variability like the MJO (for which a full cycle typically occurs in 30-60 days). In its current state the system is better suited to simulate the impact of longer term variability cycles like ENSO or the AMO.

- Line 528: while I understand it can be hard to calculate the return period of the most extreme event in a 44-year period, and that you therefore decided to place it at the 44-yr return period, I do think it's good to acknowledge that these events realistically do not have a 44-year return period but likely a way higher return period. You do somewhat say this by saying "Quantifying the severity of rare extreme events from such a shore record of observation years is an obvious issue" but I would be a bit more explicit here.

This is a great point and we have added a comment to the main text to acknowledge it (l560-l561)

- Figure 13, caption: I would mention the 100km radius in the caption as well, not just in the running text

done

- Is the model going to be open-access or can (parts of) the code be accessed anywhere? I couldn't find anything on this in the text but I might have overlooked it.

The model is not on open-access, but we do share selected datasets for academic use cases. We currently have such agreement with a student at ETH Zurich focussing on future typhoon induced rainfall risk in the Philippines.